# Epilepsy Diagnosis When the Routine Ancillary Tests Are Normal

**DOI:** 10.3390/neurolint17050066

**Published:** 2025-04-24

**Authors:** Boulenouar Mesraoua, Bassel Abou-Khalil, Bernhard Schuknecht, Hassan Al Hail, Musab Ali, Majd A. AbuAlrob, Khaled Zammar, Ali A. Asadi-Pooya

**Affiliations:** 1Neurosciences Department, Hamad Medical Corporation, Doha P.O. Box 3050, Qatar; alhail@hamad.qa (H.A.H.); mali40@hamad.qa (M.A.); majdaiabualrob72@gmail.com (M.A.A.); kzammar@hamad.qa (K.Z.); 2Weill Cornell Medical College, Doha P.O. Box 3050, Qatar; 3Department of Neurology, Vanderbilt University, Nashville, TN 2301, USA; bassel.abou-khalil@vumc.org; 4Medical Radiological Institute, 8001 Zurich, Switzerland; bschuknecht@mri-roentgen.ch; 5Department of Neurology, Shiraz University of Medical Sciences, Shiraz P.O. Box 71348-14336, Iran; aliasadipooya@yahoo.com; 6Department of Neurology, Thomas Jefferson University, Philadelphia, PA 4201, USA

**Keywords:** epilepsy, seizure, electroencephalogram (EEG), interictal epileptiform discharges (IEDs), EEG measures to improve EEG yield, magnetic resonance imaging (MRI), harmonized neuroimaging of epilepsy structural sequences MRI (HARNESS-MRI)

## Abstract

**Background/Objectives:** In a patient suspected of having epilepsy, routine EEG primarily contributes to the recording of interictal epileptiform discharges (IEDs). Similarly, magnetic resonance imaging (MRI) has become the gold standard imaging technique for identifying epileptogenic structural brain abnormalities. Various EEG and MRI tools to improve epilepsy diagnosis will be presented. **Methods:** When the initial EEG fails to record IEDs, various EEG measures that can improve EEG performance are presented; a comprehensive epilepsy-targeted MRI protocol to identify, localize, and characterize an epileptogenic lesion will also be described. **Results:** Studies show that the initial routine EEG fails to record IEDs in approximately 47–50% of epileptic patients. To improve the yield of EEG, subsequent EEG recording should include sleep deprivation, sleep recording, prolonged hyperventilation, optimized light stimulation, addition of an inferior temporal electrode chain, extended EEG duration, and continuous video-EEG monitoring, all measures known to activate IEDs. Furthermore, MRI is interpreted as “normal” in many epilepsy patients, even when performed according to an epilepsy-specific protocol and evaluated by a specialized MRI reader. In such case, the use of the Harmonized Epilepsy Structural Sequence Imaging (HARNESS-MRI) protocol and other imaging tools will improve the detection of potential epileptic lesions, as described in this study. **Conclusions:** In a patient with a clinical diagnosis of epilepsy but a normal EEG and brain MRI, several options can improve the performance of subsequent EEG and MRI examinations, the subjects of this review.

## 1. Introduction

The electroencephalogram (EEG) is the most direct measure of electrical brain activity and the most specific test for the evaluation of epilepsy [1]. Patients most commonly receive a 20 min routine EEG recorded from 19 scalp electrodes placed according to the 10–20 electrode placement. The routine EEG should include hyperventilation and intermittent photic stimulation. Ideally, waking, drowsiness, and sleep should be recorded. However, many individuals will not fall asleep during a 20 min recording. The routine EEG is not expected to capture seizures, with the exception of generalized absence seizures, which are easily precipitated by hyperventilation in the untreated patient. In a patient with epilepsy, the main input of the routine EEG is to record interictal epileptiform discharges (IEDs), which include spikes, sharp waves, spike-and-wave discharges, and polyspike-and-wave discharges. In conjunction with the clinical history, the distribution of IEDs can help classify epilepsy as focal or generalized, and can occasionally help diagnose the specific epileptic syndrome. The classification of epilepsy and epileptic syndrome may have important therapeutic and prognostic implications. Rarely, the EEG may provide evidence for the etiology of epilepsy, such as with the recording of rhythmic IEDs in patients with focal cortical dysplasia [2].

A variety of normal variants are often mistaken for IEDs, and misinterpretation of the EEG is one of the most common reasons for overdiagnosis of epilepsy. Criteria that help with the identification of discharges as epileptiform include high voltage, a duration of 20–70 ms for spikes and 70–200 ms for sharp waves, an asymmetric biphasic or polyphasic morphology with a predominant negative component, and a subsequent slow wave. In addition, an epileptiform discharge should be different from the activity around it, thus disrupting the background.

The first routine EEG fails to record IEDs in about 47–50% of patients with epilepsy [3,4]. When the clinical history is strongly suggestive of epilepsy, but the first EEG fails to record IEDs, a variety of measures can improve the yield of the EEG, which is the focus of this review.

In addition, neuroimaging is the cornerstone for etiologic assessment of patients with epilepsy. Brain magnetic resonance imaging (MRI) has evolved as the imaging technique of choice for the identification of structural brain abnormalities. However, in approximately 35% of patients with therapy refractory focal epilepsy, the MRI will be considered “normal” even when performed using an epilepsy-specific protocol and when evaluated by a specialized reader [5]. False attribution of the label “nonlesional” to an MR examination has considerable impact on the subsequent assessment and management of patients with epilepsy. A negative brain MRI reduces the probability of a patient with focal epilepsy

being referred to an epilepsy center,being recommended for epilepsy surgery,becoming seizure free following surgery.

In the following sections, we will describe a step-by-step assessment procedure for epilepsy patients with a negative MRI.

## 2. Methods

The focus of the current review is to discuss way(s) to improve the yield of EEGs and MRIs of the head by using various electroencephalographic and MRI measures in patients for whom the first EEG and first MRI were normal and in whom the clinical history strongly suggests epilepsy. Prior to compilation of this article, a literature search was conducted in Pubmed in July–August–September 2024 for English-language publications. For the section “Electroencephalogram (EEG) to Confirm Clinical Epilepsy Diagnosis—Options When the First EEG Is Normal” we searched for the terms:

“Epilepsy” or “seizures” combined with “EEG yield”, “EEG performance”, ”repeat EEG” “EEG duration”, “Sleep EEG”, “EEG sleep deprivation”, ”EEG sleep induction”, ”timing of EEG recording”, “EEG extra electrodes”, ”EEG activating procedures”, “EEG photic stimulation”, ”EEG Hyperventilation”, “ EEG Reflex precipitation of seizures”, ”EEG activating medications”, “chloralhydrate”, “secobarbital”, “temazepam”, “Chlorpromazine”, and ”clonidine“. For the section “Normal MRI in a Patient with Epilepsy—How to Increase the Yield?”, we searched for the terms: “Epilepsy” or “seizures” combined with the following:

“Standardized epilepsy protocol”, “Harmonized Neuroimaging of Epilepsy Structural Sequences” = “HARNESS-MRI protocol”, “3T MRI”, “7T MRI”, “MR field strength”, “FLAIR/T2 w sequences”, “3D Magnetization prepared rapid gradient echo = MP RAGE sequences”, “susceptibility weighted imaging = SWI”, “Diffusion weighted imaging = DWI”, and “intravenous Gd contrast administration”. Further key words searched for were targeted towards brain segmentation, postprocessing, and MR perfusion by using the term “Epilepsy” in combination with “brain MR segmentation”, “MR postprocessing by morphometric analysis”, “postprocessing by quantitative analysis of signal intensity” and “postprocessing by volumetry“, “postprocessing by 3D surface rendering technique”, “MR Fingerprinting”, and “MR perfusion”.

We also reviewed references cited in relevant publications. The articles cited are those that we considered of particular value for the purpose of this work. Whenever appropriate, we prioritized meta-analyses (not too many, unfortunately!) and research published in the last 10 to 15 years.

## 3. Electroencephalogram (EEG) to Confirm Clinical Epilepsy Diagnosis—Options When the First EEG Is Normal

### 3.1. Repeat EEG Recordings, Longer EEG Duration

The EEG yield in patients with epilepsy can be improved with repeated/serial EEG recordings or with increased EEG duration. In 429 adult patients with epilepsy (focal in 95%), the capture of IEDs increased from 50% in the first EEG to 84% by the third and 92% by the fourth EEG [4]. There was little increase in yield with additional recordings. A similar retrospective study in 1078 EEGs of 373 patients reported epileptiform discharges in 38% of patients in the first EEG [6]. The yield increased to 77% with repeated recordings, but there was little gain after the 5th recording. Among subjects with a single unprovoked seizure, the yield was 39% after the first EEG study and 68% after the third [3].

The EEG yield can also be increased by prolonging the duration of an EEG. In a prospective observational study, 426 patients had IEDs recorded during an EEG that averaged 59.4 min [7]. IEDs in 81 (19.1%) occurred only after the initial 30 min of recording. Extending the EEG from 30 min to 45 min or longer resulted in newly seen IEDs once in every 13 studies when the pretest probability of epilepsy was high [7]. Another study retrospectively evaluated the yield of the first 20 min and the subsequent 4 h of an EEG video study in 179 consecutive patients [8]. The initial 20 min EEG was nondiagnostic for 130 patients (76%). Among these patients, the subsequent 4 h of EEG video were diagnostic in 41 patients (32%). The greatest yield of the additional recording was seen in patients with focal epilepsy: the initial 20 min EEG recorded focal epileptiform abnormalities in 13 patients and the subsequent 4 h yielded epileptiform abnormalities in an additional 24 patients. For generalized epilepsy, the initial 20 min recording helped support the diagnosis in 24 patients, and the subsequent 4 h supported the diagnosis in an additional 2 patients. The additional 4 h also helped with the diagnosis of nonepileptic events. These events were captured in the first 20 min in 12 patients; the subsequent 4 h captured a nonepileptic event in 18 additional patients [8].

Several studies have examined the latency to the first interictal epileptiform discharge in outpatient EEGs. In a study of 45 adult EEGs lasting at least 60 min (mean duration 187 min) and containing an IED, the initial IED occurred in the first 20 min in 53% and after 20 min in 47% of EEGs [9]. The mean time to the first IED was 32.8 min. The latency was longer in patients with temporal IEDs (56 min) than in patients with generalized IEDs (22 min). It should be noted that the 45 EEGs in this study represented only 26% of the extended EEGs that were evaluated; no IEDs were noted in the remaining 74% [9]. Another study recorded 2–24 h EEG in 200 mixed-age patients with epilepsy [10]. IEDs were detected in 45% after 20 min, 55% after 1 h, 64% after 2 h, and 85% after 24 h. When longer duration EEG studies were evaluated, the latency to the first IED was longer. In a study of 4-day ambulatory EEG recordings in 180 adult patients with epilepsy, focal IEDs were seen in 130 and generalized IEDs in 50 patients [11]. IEDs were recorded in 44% of patients within 4 h, 58% within 8 h, 85% within 24 h, and 95% within 48 h of recording. The median latency to the first IED in the whole group was 316 min. In agreement with the prior study, IED latency was longer in focal epilepsy than in generalized epilepsy (576 vs. 43 min). Interestingly, a longer duration of epilepsy was associated with a longer latency to the first IED. Since the last 2 days of recording only added 5% new IED capture, the authors suggested that a 48 h recording was sufficient for classification in the majority of subjects [11]. In a study that evaluated 24 h EEG recordings after a first unprovoked seizure in 25 patients aged 15–59 years, 11 patients had IEDs, and these appeared after one hour of recording in 73%. The median latency to the first IED was 170 min [12].

### 3.2. Sleep, Sleep Deprivation and Sleep Induction

Studies have consistently demonstrated that recording sleep is important to capture epileptiform discharges, particularly in focal epilepsy. One study of 129 patients aged 11–79 years with a normal waking EEG found that sleep EEG detected an abnormality in 41.8% [13]. The yield of IEDs in sleep was greater with younger age [14]. A pediatric study that stratified patients by referral diagnosis reported that if the EEG while waking failed to show any IEDs, 31% of sleep studies disclosed an additional yield [15]. The additional yield of the sleep studies was notably higher in patients referred for suspected benign epilepsy with centrotemporal spikes (BECTS—now called SELECTS) and patients referred for suspected electrical status epilepticus during slow-wave sleep (ESES) [15]. In a study based on inpatient EEG monitoring, 71% of patients aged ≥10 years had their first IED in sleep [16]. In older children and adults, recording sleep is more important for focal epilepsy and particularly temporal lobe epilepsy. An inpatient EEG-video study of 210 consecutive patients aged 38.6 ± 13.9 years found that 63% of first IEDs occurred during sleep in patients with focal epilepsy, while 66% occurred in waking in patients with generalized epilepsy [17]. In 90 patients with temporal lobe epilepsy, two thirds of first IEDs occurred during sleep [17].

One advantage of longer EEG recordings is that they allow sleep to be captured. There are also methods to achieve sleep in 20–30-min routine EEG recordings. Sleep was more likely to be achieved in children if hyperventilation was performed early and photic stimulation was at the end of the recording [18]. Sleep may also be induced for shorter EEG recordings through sleep deprivation or sedation. A variety of agents have been used to induce sleep during EEGs in children. Sedation success/failure rates were similar between oral chloral hydrate, oral dexmedetomidine, oral hydroxyzine hydrochloride, and oral midazolam [19]. Several studies compared melatonin with a variety of agents or sleep deprivation for inducing sleep during an EEG in children [20,21,22,23,24,25,26,27,28,29,30,31,32]. While the comparative efficacy of melatonin varied, most studies found melatonin to be well tolerated and effective in inducing sleep during an EEG recording.

Sleep deprivation is effective at inducing sleep and may activate IEDs beyond its ability to induce sleep [33,34]. In one study, patients aged 17–75 years with a clinical diagnosis of epilepsy but with a normal or inconclusive first routine EEG underwent a second recording [35]. IEDs were recorded in 5/52 (9.6%) when the second EEG was a routine EEG and in 45/199 (22.6%) when the second EEG was sleep deprived. In another study of 20 patients aged 5–51 years with epilepsy who underwent a series of EEGs including routine, sedated, and sleep deprived recordings, IEDs were recorded in 25% of routine, 50% of sedated and 80% of sleep deprivation records [36]. It has even been suggested that the first diagnostic EEG study should be sleep deprived. Eighty-five patients aged <35 years with possible new epilepsy (having experienced at least two bilateral tonic clonic seizures) underwent three EEGs in random sequence, including a routine EEG, a sleep-deprived EEG, and an EEG with drug-induced sleep [34]. Among 36 patients with generalized spike-and-wave discharges, the sensitivity of the sleep deprived EEG was 92% compared to 58% for the drug-induced sleep EEG and 44% for the routine EEG. Ten patients (28%) only demonstrated IEDs after sleep deprivation. Among 15 patients with focal discharges, the sensitivity of the sleep deprived EEG was 73% as compared to 40% for the drug-induced sleep EEG and 27% for the routine EEG. Seven patients (47%) had IEDs seen only after sleep deprivation. In this study, sleep deprivation was extreme, with no sleep and no food after 10 pm. Two patients had generalized tonic clonic seizures after sleep deprivation. More moderate sleep deprivation (for example going to bed an hour later and waking up an hour earlier) may be less likely to precipitate unwanted tonic clonic seizures.

### 3.3. Timing of EEG Recording

The timing of the EEG may influence the EEG yield. Some forms of epilepsy have a circadian distribution of seizures as well as IEDs. For example, seizures as well as IEDs are more likely in the morning in patients with juvenile myoclonic epilepsy. One study recorded standard awake EEGs consecutively at 9 a.m. and 3 p.m. in 29 patients with juvenile myoclonic epilepsy [37]. The morning EEG showed generalized IEDs and/or a photoparoxysmal response (PPR) in 20 patients (69%), 15 of whom had a normal afternoon recording. The afternoon EEG yield of IEDs and/or PPR was only 5/29 (17%) [37]. Thus, in patients with suspected juvenile myoclonic epilepsy, a diagnostic EEG should be obtained in the morning, and if the first EEG was a normal afternoon EEG, a second EEG should be a morning EEG.

There is also evidence that the sooner an EEG is obtained after a seizure, the more likely it is to record epileptiform abnormalities. In a retrospective study of 170 adult patients who received an emergency EEG after a first unprovoked seizure, IEDs were identified in 34.1% of recordings [38]. A shorter latency from seizure to EEG was associated with a higher probability of finding epileptiform discharges. The highest probability of detecting an IED was within the first 16 h after seizure onset: IEDs were detected in 52.1% of recordings performed before versus 20.2% of those performed after that cut-off [38]. If the first EEG is normal and a second seizure occurs, then the second EEG should be obtained within 16 h after the seizure.

### 3.4. Extra Electrodes Outside 10–20 Electrode Array

Standard EEGs were usually performed using the 19 electrodes of the 10–20 electrode system. However, the 10–20 electrode array does not cover the basal aspect of the temporal lobes. Adding extra temporal electrodes may be helpful for improving temporal IED detection. Nasopharyngeal electrodes were used in the past and fell out of favor, although a recent study advocated their reconsideration [39]. Sphenoidal electrodes improved the detection of temporal IEDs [40,41,42] but their insertion can be painful, and they are not appropriate for short-term outpatient EEG studies. The true anterior temporal electrodes which are positioned over the anterior temporal region also improved the detection of temporal IEDs [43].

More recently, a study investigated the additional yield of the six inferior temporal chain electrodes of the 10–10 electrode system, F9/10, T9/10, and P9/10 [44]. Out of 70 patients with temporal IEDs, 54% of IEDs had a peak negativity in the inferior chain, and 4% were only seen at the inferior chain [44]. The inferior temporal chain is now included in the standardized EEG array of the International Federation of Clinical Neurophysiology [45].

### 3.5. Interventions with Activation Procedures

Hyperventilation and intermittent photic stimulation are standard activation procedures in the routine EEG. Hyperventilation, usually for 3 min, is known to activate generalized absence seizures in untreated children with childhood absence epilepsy. One study suggested that generalized absence seizures are more likely to be precipitated when hyperventilation is conducted in a sitting rather than supine position [46]. The yield of hyperventilation can also be increased by prolonging its duration to 5 min [47]. In a study of 877 mixed-age patients who hyperventilated for 5 min, 16% of seizures and 30% of interictal EEG abnormalities triggered by hyperventilation occurred during the last 2 min of hyperventilation, supporting the usefulness of prolonging hyperventilation for 5 min [47].

Intermittent photic stimulation can also be optimized to improve its yield as well as its safety [48,49]. In patients with suspected photosensitivity, intermittent photic stimulation should ideally be performed with three eye conditions: eyes closed, eyes open, and active eye closure (closing the eyes at the beginning of each stimulation). Active eye closure is often the most potent stimulus; it should be the default eye condition if only one eye condition is used. As mentioned above under timing of EEG, the yield of photic stimulation may also be greater in the morning [37].

When reflex precipitation of seizures is reported, the stimulus may activate both seizures and IEDs. Every effort should be made to replicate the precipitating event during the EEG. This should be quite feasible when the reported precipitating trigger is flashing light (photosensitivity), patterns (pattern sensitivity), certain types of music, reading, eating, somatosensory or proprioceptive stimuli, or startle.

In patients with suspected idiopathic generalized epilepsy, neuropsychological/cognitive tasks may activate IEDs in some patients while inhibiting IEDs in others [50,51,52]. They may be considered when the baseline EEG is normal. Cognitive tasks precipitated IEDs in 18 of 172 (6.6%) patients without abnormalities on awake EEG [51]. Among patients with activation of IEDs by cognitive tasks, 95% had idiopathic generalized epilepsy, most commonly juvenile myoclonic epilepsy followed by juvenile absence epilepsy and epilepsy with tonic clonic seizures alone [51]. In a study of 50 patients with juvenile myoclonic epilepsy, IEDs were exclusively provoked by cognitive tasks in 4 of 50 (8%) patients [53]. The yield may be even higher in subjects with known reflex precipitation of seizures by cognitive tasks. In a study of 35 patients with epilepsy who reported cognitive functions as a reflex seizure stimulus, verbal and arithmetic tasks were administered during a standard awake EEG. IEDs were activated by a verbal task in 11.4% of patients and by an arithmetic task in 5.7% [54]. All activated patients had idiopathic generalized epilepsy. Among 18 patients with normal baseline EEG, one patient demonstrated activation with both verbal and arithmetic tasks.

### 3.6. Activating Medications

As mentioned above, hypnotics administered during EEGs to induce sleep include chloralhydrate, secobarbital, and temazepam. The administration of sedation has declined considerably in favor of sleep deprivation to induce sleep, partly because conscious sedation now requires monitoring of breathing and blood pressure by nursing staff. Melatonin administration in children is exempt from this requirement.

Other medications were reported to activate IEDs when administered during an EEG. Intramuscular chlorpromazine, 50 mg, was administered to 41 adult patients with epilepsy and a normal interictal awake EEG [55]. These patients also received a separate EEG with sleep deprivation. Chlorpromazine activated IEDs in 19/20 (95%) untreated patients and 13/21 (62%) treated patients. It was superior to sleep deprivation in untreated patients, activating IEDs in seven patients who did not have IEDs on the sleep deprived EEG. All IEDs were recorded just before or after sleep onset. However, the superiority of chlorpromazine suggested an activating effect beyond its effect on sleep. There was no difference between chlorpromazine and sleep deprivation in treated patients. No IEDs were recorded with either activating procedure in 18 control subjects [55].

Clonidine is another medication used for sedation but which may also activate epileptiform activity. In a study comparing chloralhydrate and clonidine for sedation, 100 children were randomized to receive clonidine and 98 to receive chloralhydrate before undergoing EEGs. IEDs were reported in 14 children in the clonidine group and 3 children in the chloralhydrate group [56]. There was no difference between the groups in reported focal IEDs. Premedication with oral clonidine followed by intravenous administration of methohexital has been used to activate IEDs in patients undergoing magnetoencephalography. Each was found to increase IEDs in the majority of patients [57].

## 4. Normal MRI in a Patient with Epilepsy—How to Increase the Yield?

The 2017 and 2022 International League Against Epilepsy (ILAE) classification and update resulted in a three-level approach by classifying disease into seizure type, epilepsy type, and the presence of an epilepsy syndrome [58,59]. Clinical assessment and EEG monitoring are the mainstay of this classification.

Brain magnetic resonance imaging (MRI) is the imaging technique of choice to identify structural abnormalities. In patients with clinical seizures, lesions identified by MRI often enable etiologic classification such as residues of infection or trauma, glioneuronal tumors, and vascular causes (e.g., ischemic infarction). Microbleeds leading to iron/hemosiderin deposition in cortico-subcortical locations and superficial siderosis maybe indicative of a cavernoma or amyloid angiopathy, respectively. Autoimmune diseases such as limbic encephalitis and phakomatoses (tuberous sclerosis or Sturge–Weber syndrome) exhibit a pathognomonic MR appearance and thus allow further onconeural and neuronal surface antibodies direct investigations and genetic testing, respectively.

In approximately 35% of patients with therapy refractory focal epilepsy, the MRI will be considered “normal” even when performed according to an epilepsy specific protocol and evaluated by a specialized MRI reader [5].

MR-negative epilepsy is defined as an electro-clinically verified epilepsy without a structural lesion on 3T MRI applied with an epilepsy specific protocol.

False attribution of the label “nonlesional” to the MR examination has considerable impact on the subsequent assessment and management of patients with epilepsy.

A normal MR exam is particularly challenging for surgery when semiology and EEG scalp recording are non-localizing as well. Epilepsy surgery requires a sound indication based on an electro-clinical focus and a corresponding localization of an epileptogenic lesion. Following surgery, the odds of being seizure free were 2.7 times higher in patients with lesions related to the temporal lobe and 2.5 times higher in extratemporal locations. In other words, in patients with a “normal” MRI, the chance of seizure freedom drops from 70% to 46% [60].

Improved postsurgical seizure control in 75.0% of patients for temporal and in 61.5% for extratemporal lobe epilepsy has been reported when negative MR examinations were combined with subtraction ictal SPECT coregistered to MR imaging (SISCOM) and electric source imaging [61].

In the following sections, we will describe a step-by-step assessment of epilepsy patients with a negative MR:

### 4.1. Pathologies Likely to Be Missed on Brain MRI in Patients with Epilepsy

Postsurgical pathologic analysis of specimens of lesions missed by a negative brain MRI—even when performed using an epilepsy specific protocol—disclosed focal cortical dysplasia (FCD), hamartia, gliosis, and hippocampal sclerosis as the most common entities (Figure 1) [62].

The 2022 updated classification of FCD by the ILAE task force therefore recommends a four-layer integrated approach with the explicit inclusion of histology and molecular–genetic testing and neuroimaging (MRI) as layer 3 and with layer 4 merging into a diagnosis based on the information gathered by layers 1–3 [63].

### 4.2. A Standardized Epilepsy Specific Protocol Provides Superior Diagnostic Yield

A comprehensive epilepsy-targeted MRI protocol is the key factor to identify, localize, and characterize an epileptogenic lesion (Figure 1). The neuroimaging task force of the ILAE advises the “Harmonized Neuroimaging of Epilepsy Structural Sequences” (HARNESS-MRI) protocol that encompasses three “core” sequences: iso-tropic submillimetric FLAIR and 3D T1 images and high-resolution 2D T2 sequences to be performed soon after the first seizure [64].

In a recent report from Egypt [65] the authors describe an increase of causative lesions from 40 detected by a conventional MR protocol to 106 at high “cost-effectiveness based on the HARNESS protocol”. The impact of “access to detailed description of the electroclinical findings, including the semiology, suspected hemisphere/lobe” as recommended by the ILAE task force was evidenced by an additional rise to 131 lesions identified during interdisciplinary board meetings. Small focal lesions and detection of gliosis and hippocampal sclerosis require a dedicated epilepsy-specific protocol to be reliably depicted by MR.

In the setting of interval changes of seizure semiology and/or frequency, repetition of a dedicated MR protocol is recommended to evaluate for new or potentially previously missed structural lesions [66]. Performing MRI with a standardized epilepsy-specific protocol without or with contrast administration is in agreement with “The American College of Radiology Appropriateness Criteria” for “Variant 6: Known seizure disorder surgical candidate or surgical planning”. American College of Radiology (ACR) criteria are evidence-based guidelines and are reviewed annually by a multidisciplinary expert panel.

Despite the standardization achieved by the HARNESS protocol, the addition of new sequences such as SWI, DWI, and M2RRAGE and preferential application of 3D sequences as compared to 2D acquisition is an important step to further exploit morphological MRI at higher resolution and sensitivity.

An updated extension of the HARNESS protocol as suggested by the author BS is provided in Table 1.

### 4.3. Expert Readers Take Advantage of the Added Value of an Epilepsy Specific Protocol

In a prior study [67] and in a recent prospective evaluation of the HARNESS protocol [68] in 123 and 131 patients, the sensitivity of MRI was particularly enhanced through interpretation by experienced readers for the diagnosis of hippocampal sclerosis and focal cortical dysplasia as the previously most frequently missed entities in adults and children, respectively. The combination of a dedicated MR and expert reader assessment increased detection of hippocampal sclerosis from 18 to 45% and of focal lesions from 50 to 91% [67]. Expert readers are more likely to recognize focal hippocampal sclerosis (Figure 2) and detect associated indirect signs such as atrophy of the ipsilateral fornix and mammillary body. Observations like the transmantle sign and subtle changes to adjacent white matter in focal cortical dysplasia and periventricular nodular and subcortical band heterotopia are challenging and less likely to be missed visually by an experienced reader.

### 4.4. Increased MRI Field Strength and Receive Coils Improve Lesion Recognition Through Enhanced Signal and Spatial Resolution

In recent years, MRI has experienced substantial technological development related to an increase in the field strength of the magnet (e.g., 1.5 to 3T) and in the number of coil channels used to receive the signal from 32 to 64 channels. A factor that has increased the yield of MRI is the change of the “standard” field strength for brain MR from 1.5T to 3T. High field MR (3T) outperformed MRI at 1.5T field strength both in lesion detection and lesion characterization by a factor of 2.57 and 2.66 [69]. In a meta-analysis, the pooled estimate of the detection rate based on five studies was 18% (95% CI: 5–47%) for 3T in patients with prior negative 1T and 1.5T examinations, respectively [70].

The comparable detection rate for 7T MRI was higher with 23% (95%-CI: 17–30%) analyzed from seven studies. The authors admit that several factors of bias (patient selection, reference standard, information, and applicability) represent a limitation for impartial comparison of the true yield at 7T field strength.

A sign specific to 7T has been described related to FCD IIb, in which the T2* sequence depicted an intracortical black line in 5/6 patients that had not been visible on prior lower-field MR imaging [71]. Complete surgical removal of this black line resulted in seizure freedom in 4 of 5 patients, while incomplete resection led to failure of seizure freedom in the remaining individual.

The ready availability of 3T MRI and the advantage of 3T MR field strength has been approved specifically for the assessment of children [72] and overall in adult patients with epilepsy [5,68].

A practical tip is to increase the spatial resolution in a 3T field magnet by lowering slice thickness from 3 mm to 2–1.5 mm for T2 sequences while maintaining signal intensity. This is enabled by application of a head coil with a high number of receive coils—e.g., 64 channels in a 3T field system (Figure 3). Sensitivity for lesion detection is thus increased from 53.1% to 85.9% [72]. The most common pathologies additionally encountered in this study were FCD, nonspecific gliosis, and oligodendrogliosis.

Progressive substitution of 2D acquisition with 3D acquisition and thus lowering the slice thickness from 2–3 mm slice to 0.9–1 mm (=submillimetric) is a considerable yield-gaining strategy. This not only increases visual conspicuity for small lesions but moreover enables advanced postprocessing of 3D data.

Higher spatial resolution and superior gray–white matter discrimination facilitate the fusion of MR with ictal-interictal SPECT (SISCOM), magnetencephalography (MEG), and PET and additionally enables data transfer for neuronavigation during epilepsy surgery.

### 4.5. New MR Sequences Display Tissue Properties for Improved Visualization of Lesions

Though not constituents of the core HARNESS protocol, the addition of susceptibilityweighted imaging (SWI) and diffusion-weighted imaging (DWI) have acquired a status of indispensable sequences for the assessment of patients with epilepsy (see Table 1)

The SWI sequence depicts hemosiderin and calcification and thus renders visible cavernoma and cerebral and superficial siderosis as figureheads of cerebral amyloid angiopathy or posttraumatic or postsurgical siderosis (Figure 4).

Based on the SWI sequence, additional information was obtained in 13/59 (22%) of patients assessed by 1.5T MR [73].

A less-recognized feature of the SWI sequence is the paramagnetic effect of deoxyhemoglobin as a surrogate marker of vessel caliber. Pseudonarrowing of vessels may serve as a sign of regional hyperperfusion (Figure 5) and, thus, may indicate nonconvulsive status epilepticus [74,75].

Ictal and postictal cytotoxic edema within gray matter of the cortex, the hippocampus, or pulvinar thalami is demonstrated as local gray matter swelling and commonly presents with reversible diffusion restriction on DWI [75]. DWI therefore aids in distinguishing secondary postictal ischemic changes from primary pathologies such as gliosis, encephalitis, and long-term epilepsy-associated brain tumors (LEAT).

Cortical epileptogenic lesions are often subtle. Sequences that enhance gray–white matter contrast, such as the MP2RAGE sequence (see Figure 3b,c)—instead of the classical MPRAGE sequence—and the double inversion recovery sequence, have contributed to improved detection of the recently defined mild malformations of cortical development, of FCD, and of intracortical low-grade gliomas [76].

While sequence improvements have been directed to an increase of contrast between gray and white matter, the novel 3D edge-enhancing gradient echo (3D-EDGE) sequence [77] improves visualization of the gray–white boundary. Comparatively higher contrast between the lesion and the white matter of 98% relative to the gold standard FLAIR (19%) and the MP2RAGE sequence (17%) is particularly promising to facilitate visual recognition of FCD I and IIa.

### 4.6. Postprocessing by Morphometric Analysis Improves Focal Lesion Detection Beyond Visual Analysis

While visual analysis alone based on higher signal and spatial resolution increases the conspicuity of small pathologies, combination with postprocessing techniques (normalization, segmentation, and subtraction/division from a database of normal controls) and submission to a morphometric analysis program proved particularly beneficial to identify lesions that are beyond immediate visual recognition [78]. The morphometric maps created within the morphometry analysis program (Figure 6) encompass the MRI hallmarks of focal cortical dysplasia: abnormal extension of gray into white matter and configuration of deepened sulci, abnormal thickening of the cortical ribbon, and blurring of the gray–white matter junction. These are used to create morphometric extension, thickness, and junction maps. The “combined map” represents the maximum *z* score of the three maps for each voxel but is still finally verified by visual analysis [79].

Rapid analysis of a 3D MP2RAGE dataset in conjunction with a side-by-side comparison of FLAIR sequences increases sensitivity and reduces false-positive findings by morphometry. The generation of probability maps for lesions such as FCD in the future will further benefit from the integration of artificial neuronal networks [80].

### 4.7. Postprocessing by Quantitative Analysis of Signal Intensity, Volumetry Enhances Temporal Lesion Identification

Coronal high resolution T2 weighted sequences and 3D Flair sequences are standard constituents of an epilepsy specific protocol to render visible hippocampal sclerosis, FCD type II, and areas of gliosis. Data postprocessing by MRI quantification relates to calculation of Flair and T2 signal (relaxometry) of the hippocampus and within lesions of the temporal lobe. Volumetry signifies quantification of the volume of the hippocampus and amygdala (Figure 7).

In 78 patients with temporal lobe epilepsy, T2 relaxometry identified lesions in an additional 15 patients (19%) with visually normal MRI and volumetry identified a further 10 cases (13%) of hippocampal sclerosis. The combination of T2 relaxometry and volumetry provided the highest yield with recognition of hippocampal sclerosis in 22 (28%) patients with previous normal interpretations [81]. However, contrary to hippocampal sclerosis, FCD type I lesions, despite postprocessing, still have a high association with MR interpretation as “nonlesional”.

### 4.8. Postprocessing by 3D Surface Rendering Technique to Improve Comprehension of Superficially Located Lesions

Three-dimensional surface-based reconstruction encompasses the postprocessing of cortical lesions and thus provides a visual depiction of the surface pattern of gyri and sulci (Figure 8). Submillimetric 3D acquisitions are a prerequisite to provide high spatial resolution in order to facilitate visual lesion detection and apply neuronavigation for intraoperative lesion identification.

Up to 40% of FCD II are located in the central region. A particular “pattern of the central sulcus” characterized by a focal hook-shaped configuration of the precentral gyrus intersecting the central has been described as an indication of a hidden FCD II [82]. As this sulcal pattern mimics the shape of the power symbol on electronic devices, the term “power button sign” was coined. Though this surface sign is meant to improve presurgical comprehension rather than identification, the power button sign enabled additional detection of centrally located FCD in 6/13 patients with prior negative MRI (46%) [82].

### 4.9. MR Fingerprinting Exploits Tissue Properties to Reflect “Activity” of a Lesion

MR fingerprinting is an evolving technique based on using a single scan to cover the entire brain to assess multiple tissue properties by secondarily calculating T1 and T2 values and visualizing the tissue structure in tissue fraction maps. MR fingerprinting was able to differentiate active/epileptic nodules from inactive ones in heterotopia and to identify the tubera with the highest epileptogenicity in patients with tuberous sclerosis [83].

### 4.10. MR Perfusion Exploits Blood Flow to Identify Interictal Laterality of a Focus

Accurately localizing the epileptogenic focus preoperatively in a nonlesional MR is the key to surgical success. MRI perfusion enables insights into blood flow and volume in patients with focal seizures. Seizure activity leads to an increase in local perfusion (Figure 5b). Perfusion MR conducted with intravenous contrast administration is based on the dynamic susceptibility contrast (DSC) technique and may be performed without iv Gadolinium contrast administration by arterial spin labeling (ASL). Contrast enhanced DSC MRI is assessed by a gradient echo (GRE)–echo planar imaging (EPI) sequence during the first pass of a bolus dose of Gd (0.2 mmol/kg body weight). A comparative study between DSC and ASL non-contrast perfusion showed interictal cerebral blood flow to be significantly lower in patients with temporal lobe epilepsy according to either technique compared to the contralateral (unaffected) side. The correlation coefficients between clinical laterality and perfusion were 0.86 for ASL and 0.83 for the DSC technique [84].

PET-MR is a hybrid technique that was used to interrogate perfusion by the ASL MR technique and glucose metabolism (PET) in 20 patients with MR-negative refractory focal epilepsy [85]. Concordance of ASL and ^18^F-FDG PET were excellent for lateralization and good for localization of the epileptogenic focus.

### 4.11. MR Coregistration of SPECT and PET Aids by Localizing Focal Changes in Brain Perfusion, Glucose Metabolism or Specific Neurotransmitters

Single-photon emission computed tomography (SPECT) as well as positron emission tomography (PET) rank among the established functional imaging tests for the presurgical evaluation of the seizure onset zone. SPECT consists of technetium-99 m-labeled substances which reflect blood flow/perfusion and PET uses fluorine-18 fluorodeoxyglucose ([^18^F]FDG) as a marker of neuronal activity.

SPECT is used for localizing the epileptogenic zone based on an interictal and ictal examination. Digital subtraction of both studies complements visual inspection to distinguish seizure-related increases in blood flow from preexisting areas of decreased perfusion and hypometabolism in a corresponding location. The dose applied is about 5 mSv per acquisition. SISCOM (subtraction ictal SPECT CO-registered to MRI) enables superior anatomic localization of the tracer activity when the result of interictal/ictal SPECT subtraction is fused to a 3D MR image. In a meta-analysis of 11 studies comprising 142 MRI-negative patients, the SISCOM positive rate was 83.8%. The pooled positive predictive value by concordant SISCOM was 56% in 178 surgical patients [86].

SISCOM holds potential in the identification and anatomic localization of seizure foci and thus plays an important presurgical role.

Coregistration of FDG-PET and MRI is an alternative used for the localization of the epileptogenic zone in patients with drug-resistant epilepsy. In case of questionable MR findings or a presumably negative MRI result, PET performed in the seizure-free interval is aimed at the identification of cerebral regions with decreased glucose metabolism. Using EEG or surgical outcome as the gold stand, a metanalysis of 44 studies resulted in concordance rates of 67% for FDG-PET as a sole study and of 0.93% when FDG-PET/MRI coregistration was performed [87]. PET/MRI coregistration is a promising strategy to guide intracranial electrode placement and an unequivocal finding to abstain from invasive EEG recording.

### 4.12. Contrast Administration Is of Limited Gain with Respect to Lesion Detection but May Contribute to Characterization of an Abnormality

Intravenous administration of Gd-based contrast agents in patients with epilepsy provides a low additional yield. In a study of 150 patients with focal epilepsy, contrast uptake was present in 33 out of 69 patients (47.8%) [88]. In fact, contrast enhancement only revealed two additional lesions (6.0%) that had passed unrecognized on nonenhanced sequences. The presence of contrast enhancement allowed for the characterization of lesions to better advantage and altered the diagnosis in 17 of 69 patients (24.6%). In patients who harbor a long-term epilepsy-associated tumour (LEAT) or patients with increased frequency of seizures, contrast administration may indicate a conversion of the dignity of a preexisting lesion (Figure 9) or a de novo abnormality.

In a recent study on the utility of iv Gd administration in MRI performed for the evaluation of acute-onset pediatric seizures, only 5% of cases retrospectively necessitated the use of contrast agent [89]. Sixty-nine percent of these lesions were classified as neoplastic, while thirty-one percent had an immune/infectious etiology. The authors esteem the use of intravenous Gd administration as being of “limited additive benefit” in most cases.

The expert panel on neurological imaging [66] in the “ACR Appropriateness Criteria^®^ Seizures and Epilepsy” recommends “MRI head without and with IV contrast or MRI head without IV contrast” as being an appropriate and equivalent presurgical imaging approach for patients with known epilepsy.

### 4.13. New Entities: Knowledge Shapes Perception

Advances in MR imaging technique, postprocessing, and the growing expertise of readers potentially reduce the number of normal MR examinations in patients with epilepsy. However, as “we see what we know”, learning and identification of previously unknown or unrecognized pathologies play an important role in imaging interpretation.

Anterior temporal encephaloceles are lesions defined as brain herniation through small—commonly antero-inferior—middle cranial fossa skull base defects in patients with temporal lobe epilepsy. These encephaloceles are recognized by scrutinizing the cortex of the anterior temporal lobe on coronal T2/Flair MR sequences. When identified and resected, anterior temporal encephaloceles had an excellent postsurgical outcome defined as ILAE Class 1 or 2 achieved in 85% of 20 patients [90]. The majority of these encephaloceles (20/37, 54%), however, had previously passed unrecognized on initial image interpretation even by experienced readers (Figure 10).

It is important to keep in mind that temporal meningo-encephaloceles have to be interpreted in the clinical context and in conjunction with EEG recordings, as this entity can also be present in subjects without epilepsy.

An additional entity that has increasingly attracted attention is nontumoral amygdala enlargement (Figure 11). Amygdala enlargement has been reported in 12% of patients with mesial temporal lobe epilepsy (MTLE) and was detected by volumetry and T2 relaxometry in patients without visible MRI abnormalities [91]. The etiology may be heterogenous: secondary and often reversible on MR in patients with adequate antiseizure medications response but may be termed “primary” based on amygdala dysplasia in those patients with an MR showing irreversible amygdala enlargement.

A continuing deficit in MR imaging identification of subtle lesions holds true for focal cortical dysplasia despite well-established MR criteria. Patients who harbor an electro-clinically identified focus are a promising group for re-evaluation of MRI. Adding a hypothesis about focus localization was a simple means to increase visual detection of a focal cortical dysplasia from 22.7% to 36.4% [92].

Recent histopathologic developments as well as the results of molecular and genetic testing play an increasing role in the assessment of FCD, particularly type II. It is essential, therefore, to follow these new observations and translate histologic and molecular testing results into imaging interpretation as indicated in the four-layer approach mentioned earlier on.

New molecular–genetic observations comprise mTOR pathway activation that defines FCD type II, hemimegalencephaly, and tubera of tuberous sclerosis as a continuum of a shared genetic variant [63]

A newly described entity is MOGHE, mild malformations of cortical development with oligodendroglial hyperplasia and epilepsy [93]. The lesions on MR are predominantly frontal in children and show two subtypes: linear cortico-subcortical FLAIR and T2 hyperintensity and reduced cortico-medullary differentiation. *SLC35A2* somatic mutations have recently been identified [94]. MOGHE is an excellent example of the four-layer diagnostic approach integrating electro-clinical presentation, histopathology, imaging findings, and genetic analysis [63].

Postsurgical review of brain MR in 12 patients with MOGHE disclosed findings similar to FCDIIa [95] such as focal mild cortical thickening in predominantly frontal 50% and temporal and multilobar 25% locations, gray–white matter junction blurring on T2/FLAIR sequences, and a depression of the cortical gyrus with resultant widening of sulcus (cortical dimple) on 3D surface reconstructions.

Differentiating lesions that are sequela of a seizure or seizure status rather than causative maybe a challenge. A systematic review of 11 publications by Mariajoseph and colleagues [75] lists cortico-subcortical T2/FLAIR signal changes commonly with diffusion restriction, adjacent leptomenigeal Gd enhancement, and involvement of the hippocampus and splenium of corpus callosum due to bright T2/FLAIR signal as the most frequent observations. Lesions tend to regress within a variable time interval as early as 5 days but may display prolonged resolution following status epilepticus [96].

Contrary to the bright T2/FLAIR signal mentioned above, a low T2 white matter signal leading to changes in the so-called “dark white matter” sign has gained little attention in the context of epilepsy. In a systematic review on low T2/FLAIR white matter signal, this sign was found to be indicative of nonketotic hyperosmolar hyperglycemia in 51.4% of patients with seizures [97] The dark white matter sign is commonly reversible when related to seizures but is not specific for epilepsy. This sign was also encountered in patients suffering from encephalitis, Moya Moya disease, or subdural hematoma.

## 5. Conclusions

In a patient with a clinical diagnosis of epilepsy but a normal EEG, there are a number of options to improve the yield of the next EEG study (Table 2). The next EEG recording should ideally include measures known to activate IEDs or optimize IED capture, including sleep deprivation, recording of sleep, prolonged hyperventilation to 5 min, optimized photic stimulation, addition of inferior temporal electrode chain, prolonged EEG duration, and continuous video-EEG monitoring. If juvenile myoclonic epilepsy is suspected, then the next EEG should be a morning EEG, with most of the recording after arousal from sleep. If reflex precipitation is reported, then there should be an attempt to replicate the precipitating stimulus during the recording. If the patient is having recurrent seizure events with a cyclical pattern, then the next recording should be scheduled to coincide with the expected timing of events, and if not feasible, then it should be arranged as soon as possible after the event, preferably within 16 h.

Regarding neuroimaging, a nonlesional MR examination is a considerable obstacle for patients with epilepsy to gain a satisfactory postsurgical result (ILAE 1-2). Morphological MRI performed following an epilepsy specific protocol and interpreted by an expert reader is the mainstay of structural assessment of patients with focal epilepsy. MRI interpretation is best accomplished on a 3T magnet with a 64 channel receive head coil with knowledge of semiology, a focus hypothesis, and EEG findings.

The core HARNESS protocol has to be supplemented by diffusion-weighted imaging and susceptibility-weighted and 3D MP2RAGE sequences (Table 1).

Advanced techniques such as morphometric analysis, quantitative MR assessment based on T2 and FLAIR signal intensity, and volumetry contribute to the increased detection of lesions that may be very subtle and therefore visually inconspicuous, such as hippocampal sclerosis, malformations of cortical development, migrational disorders, or dual pathology. Nonlesional MR should benefit from an extended MR examination (Table 1) including morphometry, quantitative postprocessing, and—in many patients—a contrast enhanced sequence.

SISCOM and PET coregistered to MR are additional valuable tools to increase the yield of a presumably normal MR examination.

A stepwise approach to visual assessment of MR images, supplemented by morphometry and postprocessing, and integration of molecular and genetic testing and histology within an interdisciplinary team rather than in radiologic isolation provides the highest diagnostic yield and a great chance to escape false designation of an MR examination as nonlesional.

## Figures and Tables

**Figure 1 neurolint-17-00066-f001:**
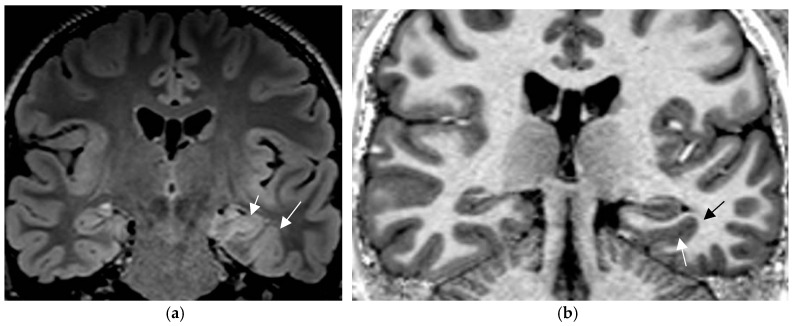
Histologically confirmed FCD I depicted by abnormal deep and straight left. collateral sulcus with a slight blurring of the gray–white matter junction on the FLAIR image. (**a**) (1.7 mm) and signal hyperintensity of the cortical border. Reduction of adjacent parahippocampal white matter (white arrow on MP2RAGE sequence) (**b**) slice thickness 0.9 mm. The hippocampus displays a mild signal increase on the FLAIR image (without atrophy) as a sign of dual pathology. Both lesions were missed on prior 1.5 T examination in a patient with left temporal lobe epilepsy.

**Figure 2 neurolint-17-00066-f002:**
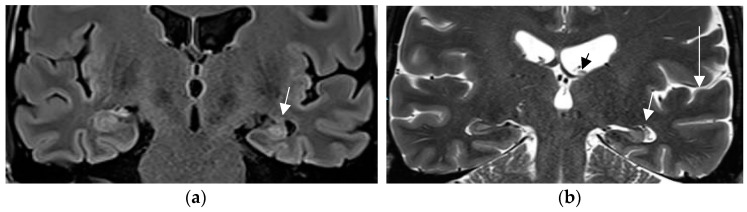
Hippocampal sclerosis depicted on coronal FLAIR (**a**) and T2 w sequence (white arrows) (**b**) (1.7 mm each) with volume reduction of the CA1-4 regions of the left hippocampus. FLAIR and T2 signal increases and slight atrophy of the left fornix (short arrow in (**b**)). Marked volume reduction of left temporal lobe as evidenced by the lower position of the left Silvian fissure (long arrow in (**b**)) compared to the right side.

**Figure 3 neurolint-17-00066-f003:**
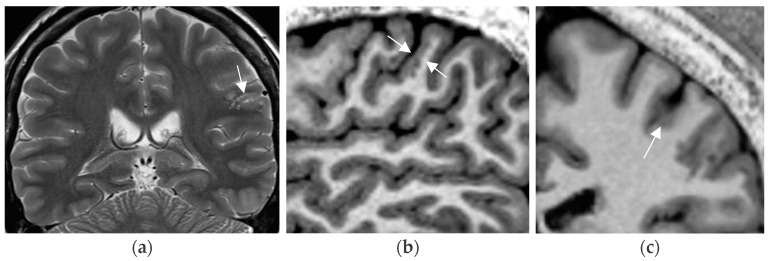
Mild malformation of cortical development along left central sulcus hardly depicted by slight cortical blurring and subcortical gliotic foci on 1.7 mm coronal T2 w image (arrow in (**a**)) (**a**). Improved visualization of the mMCD on sagittal (**b**) and coronal (**c**) MP2RAGE sequence (0.9 mm) with slight nodular irregularity of both cortical borders and focal subcortical extension (**c**).

**Figure 4 neurolint-17-00066-f004:**
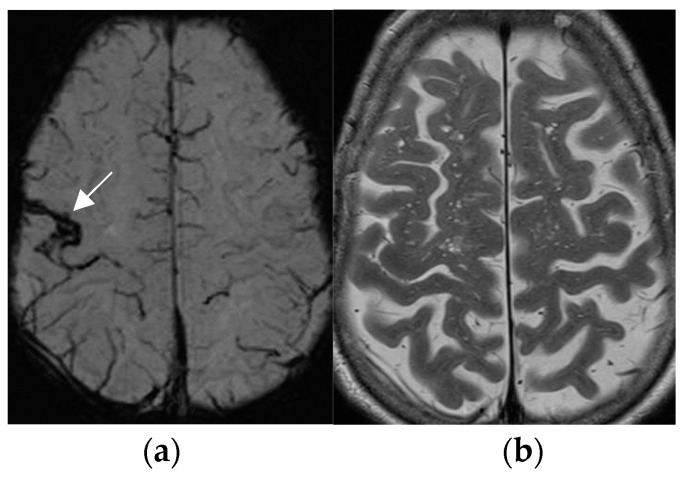
Amyloid angiopathy as evidenced by superficial siderosis within the right central sulcus on SWI sequence (arrow in (**a**)); invisible on the corresponding axial T2 w image (**b**); patient with focal seizures localized within the left hand.

**Figure 5 neurolint-17-00066-f005:**
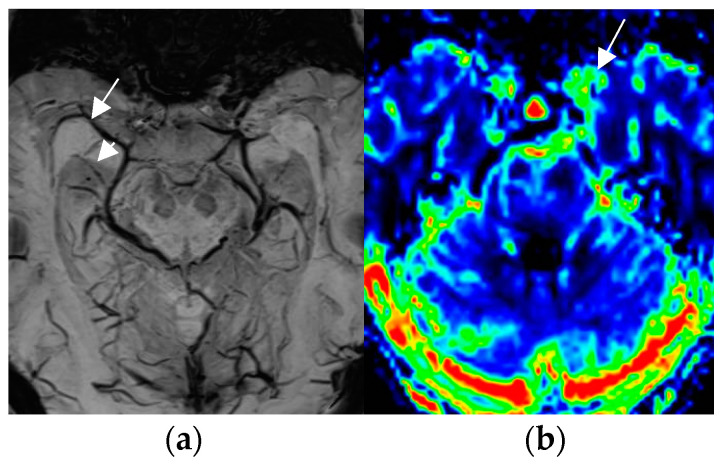
Non-convulsive absence status with left-sided pseudonarrowing of the mesiotemporal portion of basal vein (arrow in (**a**)) and hippocampal veins (arrowhead in (**a**)) as compared to the normal right side and mesiotemporal hyperperfusion; (CBF = cerebral blood flow image (**b**)) indicated by left light-green mesiotemporal hypervascularity (arrow in (**b**)).

**Figure 6 neurolint-17-00066-f006:**
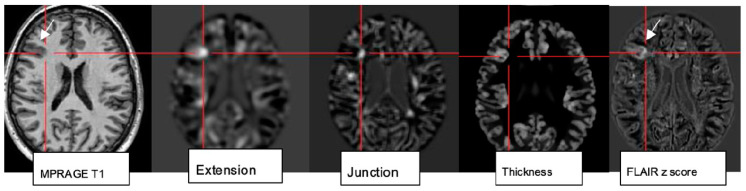
Morphometric analysis by normalization, segmentation, and subtraction/division from a database of healthy controls results in the calculation of z score maps: extension corresponds to abnormal gyration/sulcation, location of gray matter (GM); junction signifies blurring of GM-WM border; and “thickness” depicting cortical thickening in a patient with right frontal FCDIIb (see MPRAGE T1, FLAIR z score image).

**Figure 7 neurolint-17-00066-f007:**
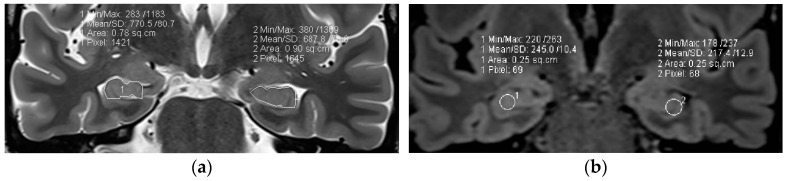
MR volumetry (**a**) and T2/FLAIR relaxometry (**b**): right hippocampal abnormality histologically verified as hippocampal sclerosis and gliosis as evidenced by a mild area decrease of the right hippocampus (0.78 cm^2^ versus 0.90 cm^2^) and a slight FLAIR signal elevation (245 vs. 217). Both findings were suspected visually but were verified by postprocessing of volume and signal intensity.

**Figure 8 neurolint-17-00066-f008:**
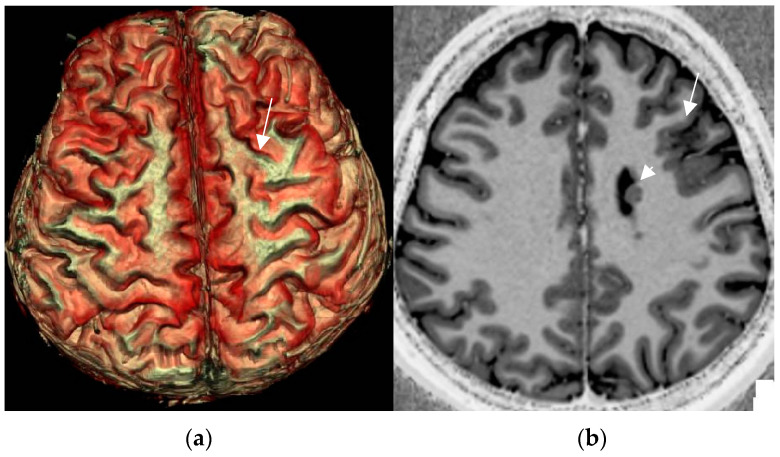
3D surface-based reconstruction of cortical folds in left frontal FCD II depicted by a cortical dimple (arrow) on 3D surface-based reconstruction (**a**). The underlying cause is a left frontal FCD (arrow) in an unusual combination with periventricular nodular heterotopia (arrowhead) depicted on the MP2RAGE sequence (**b**).

**Figure 9 neurolint-17-00066-f009:**
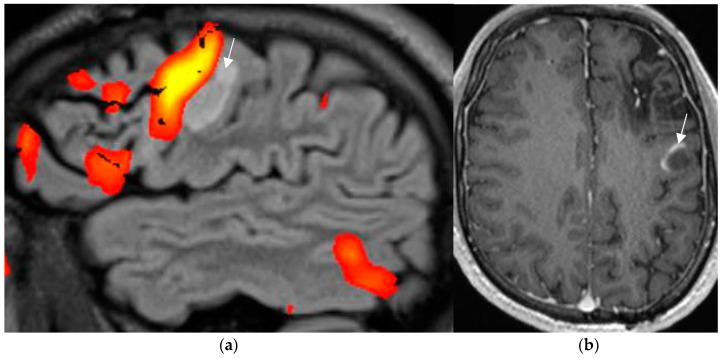
Left Precentral tuber on a sagittal FLAIR image ((**a**)—arrow) in a patient with known tuberous sclerosis. Recent-onset right focal motor seizures and transient speech arrest lead to a functional MRI. (**a**) Left speech dominance and precentral hand activation during motor task. New Gd contrast uptake is present on the axial MPRAGE Gd-enhanced sequence (**b**) in the periphery of the left precentral “active” tuber (arrow in (**b**)).

**Figure 10 neurolint-17-00066-f010:**
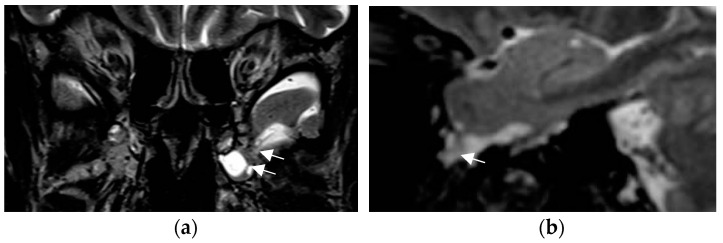
Temporal meningoencephalocele depicted by herniation of the left temporal pole into a small anterior skull base defect with resultant CSF entrapment below the middle cranial fossa floor (a: coronal T2 w image 1.7 mm). A thin stalk of gliotic tissue connects to the left temporal pole (top arrow in coronal T2 (**a**) and arrow in sagittal FLAIR image 0.9 mm (**b**).

**Figure 11 neurolint-17-00066-f011:**
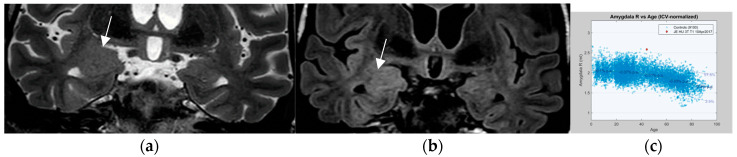
Nontumoral right amygdala enlargement on a coronal T2 (1.7 mm arrow in (**a**)) and FLAIR sequence (0.9 mm) with signal elevation on the coronal FLAIR image (arrow in (**b**)) visually increased volume is confirmed to exceed 2nd standard deviation of normal value on volumetry (**c**).

**Table 1 neurolint-17-00066-t001:** Updated extended HARNESS epilepsy-specific protocol.

A: Core Sequences	B: Additional Sequences with Evidence of Gain
isotropic sagittal ≤ 1 mm 3D T1 sequ. eg MPRAGE	isotropic sagittal ≤ 1 mm 3D T1 IR sequ. eg MP2RAGE (may replace MPRAGE)
isotropic sagittal ≤ 1 mm 3D FLAIR sequ. no interslice gap	axial hemosiderin sensitive susceptability weighted sequ. SWI displayed as MIP
high resolution (e.g., 0.4 × 0.4 × < 2 mm) coronal 2D T2 images	diffusion weighted sequ. DWI 2.5–3 mm when deemed appropriate by change of semiology/seizure frequency ± MR lesion:
	isotropic ≤ 1 mm 3D T1 MPRAGE Gd enhanced (0.2 mmol/KG)

**Table 2 neurolint-17-00066-t002:** Measures to activate IEDs in patients with epilepsy.

Measure	Comment
Repeat routine EEG	No added benefit after the 4th recording
Longer EEG recording	Most helpful for focal epilepsy 48 h usually sufficient for classification
Include sleep	Particularly important for focal epilepsy
Sleep deprivation	Particularly high sensitivity in generalized epilepsy Should not be extreme, to avoid precipitation of convulsive seizures
Optimize timing of EEG	Morning EEG has higher sensitivity for juvenile myoclonic epilepsy EEG yield is higher if performed within 16 h of last seizure
Extra electrodes	Inferior temporal chain may improve EEG yield in temporal lobe epilepsy
Optimize hyperventilation	Greater yield in sitting position Prolonging hyperventilation to 5 min may improve yield
Optimize photic stimulation	Active eye closure at onset of flash has highest sensitivity A photoparoxysmal response is more likely to be elicited in the morning
Include reflex seizure precipitants during EEG	Specific triggers should be used during EEG Cognitive tasks may activate IEDs in some patients with idiopathic generalized epilepsy

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
