# Peer review of "Epilepsy Diagnosis When the Routine Ancillary Tests Are Normal"

_2035-8377, 2025, doi:10.3390/neurolint17050066_

Round 1
Reviewer 1 Report
Comments and Suggestions for Authors
It is a smart concise review appropriately referenced and illustrated. Fairly comprehensive well put review article looking more like a book chapter that an original contribution. I would expect authors' own analysis and conclusions to make it an original contribution.
In addition to the novelty factor, the chief concern is about the conclusion section which enumerates only the most basic EEG diagnostics without mentioning continuous video EEG monitoring. The Neuroimaging paragraph in conclusions focuses on the Harness protocol but does not mention more advanced co- registration protocols etc.
Comments on the Quality of English Languagegood
Author Response
Reviewer' 1 comments
Quality of English Language
(x) The English could be improved to more clearly express the research.
We have reviewed the manuscript and made edits to improve clarity
Comments and Suggestions for Authors
It is a smart concise review appropriately referenced and illustrated. Fairly comprehensive well put review article looking more like a book chapter that an original contribution. I would expect authors' own analysis and conclusions to make it an original contribution.
The conclusion has been rewritten in order to put more emphasis on advanced MR techniques and MR co- registration
In addition to the novelty factor, the chief concern is about the conclusion section which enumerates only the most basic EEG diagnostics without mentioning continuous video EEG monitoring.
Video-EEG monitoring was added in the conclusion section
The Neuroimaging paragraph in conclusions focuses on the Harness protocol but does not mention more advanced co- registration protocols etc.
A paragraph on nuclear medicine techniques that enable coregistration to MR ( SPECT , SISCOM and PET ) has been added in the appropriate place paragraph 3.11.
Reviewer 2 Report
Comments and Suggestions for Authors
The manuscript is a nice review of what can and must be done to confirm the diagnosis of epilepsy (when correctly suspected) and how to clarify the cause of focal epilepsy (when correctly suspected by symptoms and EEG findings).The review nicely summarizes the current knowledge on the place and value of EEG.
I would only suggest mentioning the role of melatonin in sleep induction, especially in children, as this is the easiest way to perform a nap EEG with or without partial sleep deprivation, and this is a routine nowadays.
In the MRI (neuroimaging) section, I think the place of FDG-PET is not accurately presented. In my opinion, this is the most widely used and easily accessible method for examining drug-resistant epilepsy. It has its own place when used with MRI, not only when comparing ASL MR imaging to PET, but also when suspecting a lesion on MRI and co-registering it (T1, FLAIR, etc.) to the PET. This is extremely important when considering further, i.e. invasive, exploration of the patient.
Figure 5. a,b needs to be improved because the arrow in a) and the arrowhead in b) are not visible.
When describing the role of post-processing by morphometric analysis of MRI, the authors must cite and refer to the original works by Huppert HU, et al. (2005).
MOGHE is a great example of a 4-layer integrative diagnosis, so it's important to mention the landmark paper by Hartlieb T, et al. (2019) on the age-specific features of MRI lesions in children, who are usually diagnosed with this disease based on the characteristic clinical course (as described by Barba C, et al., 2023).
Author Response
Reviewer' 2 comments
Quality of English Language
(x) The English is fine and does not require any improvement.
Comments and Suggestions for Authors
The manuscript is a nice review of what can and must be done to confirm the diagnosis of epilepsy (when correctly suspected) and how to clarify the cause of focal epilepsy (when correctly suspected by symptoms and EEG findings). The review nicely summarizes the current knowledge on the place and value of EEG.
I would only suggest mentioning the role of melatonin in sleep induction, especially in children, as this is the easiest way to perform a nap EEG with or without partial sleep deprivation, and this is a routine nowadays.
We have added a section and references regarding the use of melatonin.
In the MRI (neuroimaging) section, I think the place of FDG-PET is not accurately presented. In my opinion, this is the most widely used and easily accessible method for examining drug-resistant epilepsy. It has its own place when used with MRI, not only when comparing ASL MR imaging to PET, but also when suspecting a lesion on MRI and co-registering it (T1, FLAIR, etc.) to the PET. This is extremely important when considering further, i.e. invasive, exploration of the patient.
As the mainstay of the article is to address nonlesional MR the role of FDG PET co-registered to MR – and SPECT /SISCOM has been added in an new additional paragraph: 3.11 and the value for invasive electrode placement has been mentioned
Figure 5. a,b needs to be improved because the arrow in a) and the arrowhead in b) are not visible.
Figure 5 : arrows have been reinserted.
When describing the role of post-processing by morphometric analysis of MRI, the authors must cite and refer to the original works by Huppert HU, et al. (2005).
The original article by Huppertz et al. 2005 has been added
MOGHE is a great example of a 4-layer integrative diagnosis, so it's important to mention the landmark paper by Hartlieb T, et al. (2019) on the age-specific features of MRI lesions in children, who are usually diagnosed with this disease based on the characteristic clinical course (as described by Barba C, et al., 2023).
MOGHE and the suggested papers have been introduced Hartlieb T, et al. (2019)
Reviewer 3 Report
Comments and Suggestions for Authors
- page 2 (Introduction), first sentence: add reference.
- page 2 (Introduction), line 18: when listing the criteria of an IED, add "disruption of background"
- page 2 (Methods): why was only Pubmed searched, and no other databases?
- page 3 (Methods), line 16: will the entire search strategy (including a PRISMA flow chart) be included as supplementary material to the article?
- page 10, figure 5: no arrows to be seen on the figure?
- page 12, line 17: explain "power button sign" more in detail
- page 13, figure 9: not explained what the arrows point at.
- page 14, line 16: anterior temporal encephaloceles can also be present in patients without epilepsy
- overall comment: some sentences are too long to understand and are not phrased clearly. The punctuation marks (") are not always used correctly.
Author Response
Reviewer' 3 comments
Quality of English Language
(x) The English could be improved to more clearly express the research.
We have reviewed the manuscript and made edits to improve clarity
Comments and Suggestions for Authors
- page 2 (Introduction), first sentence: add reference. Reference has been added.
- page 2 (Introduction), line 18: when listing the criteria of an IED, add “disruption of background” Added
- page 2 (Methods): why was only Pubmed searched, and no other databases? Pubmed will have the most relevant references
- page 3 (Methods), line 16: will the entire search strategy (including a PRISMA flow chart) be included as supplementary material to the article? No it will not. It was not performed
- page 10, figure 5: no arrows to be seen on the figure?
Figure 5 and 9 have been improved
- page 12, line 17: explain “power button sign” more in detail
Power button sign is explained in the manuscript
- page 13, figure 9: not explained what the arrows point at.
Done
- page 14, line 16: anterior temporal encephaloceles can also be present in patients without epilepsy
Thank you .The remark on Temporal encephalocles – absolutely correct - has been
integrated
Comments on the Quality of English Language
- overall comment: some sentences are too long to understand and are not phrased clearly. The punctuation marks (") are not always used correctly.
We have edited the punctuation marks